# Global Archaeal Diversity Revealed Through Massive Data Integration: Uncovering Just Tip of Iceberg

**DOI:** 10.3390/microorganisms13030598

**Published:** 2025-03-05

**Authors:** Antonios Kioukis, Antonio Pedro Camargo, Pavlos Pavlidis, Ioannis Iliopoulos, Nikos C Kyrpides, Ilias Lagkouvardos

**Affiliations:** 1Department of Clinical Microbiology, School of Medicine, University of Crete, 70013 Heraklion, Greece; med1p1120008@med.uoc.gr; 2DOE Joint Genome Institute, Lawrence Berkeley National Laboratory, Berkeley, CA 94720, USA; antoniop.camargo@lbl.gov (A.P.C.); nckyrpides@lbl.gov (N.C.K.); 3Foundation for Research and Technology Hellas, Institute of Computer Science, 70013 Heraklion, Greece; pavlos.pavlidis@uoc.gr; 4School of Medicine, University of Crete, 70013 Heraklion, Greece; iliop.john@gmail.com

**Keywords:** *Archaea*, genetic diversity, microbial dark matter, *Asgardarchaeota*, IMNGS

## Abstract

The domain of *Archaea* has gathered significant interest for its ecological and biotechnological potential and its role in helping us to understand the evolutionary history of *Eukaryotes*. In comparison to the bacterial domain, the number of adequately described members in *Archaea* is relatively low, with less than 1000 species described. It is not clear whether this is solely due to the cultivation difficulty of its members or, indeed, the domain is characterized by evolutionary constraints that keep the number of species relatively low. Based on molecular evidence that bypasses the difficulties of formal cultivation and characterization, several novel clades have been proposed, enabling insights into their metabolism and physiology. Given the extent of global sampling and sequencing efforts, it is now possible and meaningful to question the magnitude of global archaeal diversity based on molecular evidence. To do so, we extracted all sequences classified as *Archaea* from 500 thousand amplicon samples available in public repositories. After processing through our highly conservative pipeline, we named this comprehensive resource the ‘Global *Archaea* Diversity’ (GAD), which encompassed nearly 3 million molecular species clusters at 97% similarity, and organized it into over 500 thousand genera and nearly 100 thousand families. Saline environments have contributed the most to the novel taxa of this previously unseen diversity. The majority of those 16S rRNA gene sequence fragments were verified by matches in metagenomic datasets from IMG/M. These findings reveal a vast and previously overlooked diversity within the *Archaea*, offering insights into their ecological roles and evolutionary importance while establishing a foundation for the future study and characterization of this intriguing domain of life.

## 1. Introduction

In the early days, all non-eukaryotic single-celled microorganisms were classified as ‘Prokaryota’ based on their morphology. Advances in comparative and phylogenetic methods have clarified the distinctions between Prokaryota and the three domains of life, *Eukarya*, *Eubacteria*, and *Archaebacteria*, which were traditionally considered the three primary domains of life. Initially, *Archaebacteria* were classified as part of a single domain alongside *Bacteria*. However, subsequent research revealed significant distinctions, leading to their reclassification into two separate domains: *Archaea* and *Bacteria* [1]. The genetic and biochemical similarities (presence of histones, complex RNA polymerases, etc.) between *Eukaryotes* and *Archaea* led Carl Woese to propose that *Archaea* are closer to Eukaryotes than to Bacteria [2]. This relation is being reinforced to this day, with the discovery of novel species belonging to the new order of *Asgardarchaeota*, which contain multiple new shared characteristics between Archaea and Eukaryotes [3,4].

Despite advancements in DNA amplification techniques, including refined polymerase chain reaction (PCR) methodologies, as well as reduced sequencing costs, PCR-free sample preparation protocols, and the utilization of the 16S ribosomal RNA (rRNA) marker gene, our comprehension of microbiology remains constrained by the challenges associated with laboratory cultivation [5,6,7,8]. The selection of this marker gene is a result of its evolutionary conservation across all known life forms, which allows it to serve as a basis for phylogenetic and taxonomic identification. Initially, *Archaea* were categorized into two superphyla: *Euryarchaeota* and *Crenarchaeota* [3,9,10]. However, the expansion of sampling efforts across diverse environments has unveiled evolutionary branches that do not fit neatly into the proposed superphyla, which are exemplified by *Korarchaeota* [11]. Furthermore, even within these emerging superphyla, deeply branching microbial groups have been identified, prompting the coining of the term ‘microbial dark matter’ [12] to describe this uncultivated diversity.

The number of *Archaea* phyla has steadily increased in the last decade, now reaching twenty [13]. The discovery of these new phyla and other taxonomic groups is facilitated by novel techniques like metagenomic assembly and binning as well as the method of single-cell amplification and sequencing [14,15,16,17,18]. From the manual Sanger protocol to the tabletop sequencers of today, the progress has been unimaginable. Those advances dropped the cost of sequencing vertically, enabling us to produce more microbial data than ever before. In conjunction with the development of new computational methods, the rate at which novel *Archaea* are being uncovered has only accelerated.

The exploration of ecological niches has shown an unequal emphasis, particularly with a predominant focus on *Bacteria*. Several factors have contributed to this skew. The paramount importance placed on identifying *Bacteria* with harmful implications for human health, such as the sequencing of *Haemophilus influenzae* in 1995, has driven research toward this direction. Likewise, attention was directed toward economically significant microorganisms, which is exemplified by the sequencing of *Saccharomyces cerevisiae* in 1996. In contrast, *Archaea* have not received comparable attention as they neither pose direct pathogenic threats to humans nor offer immediate economic benefits. The design of primers targeting the bacterial 16S rRNA gene, the relatively low abundance of *Archaea* in many microbial communities, and the challenges associated with laboratory cultivation further hinder efforts to even roughly estimate the diversity of archaeal organisms. Consequently, attempts to estimate its microbial diversity have exhibited a substantial range, varying from a few million [19] to a few billion [20].

Although it is true that all microbes contribute to carbon and nutrient cycles, certain groups, e.g., *Archaea*, possess unique metabolic pathways that distinguish their roles within these processes. Given their dispersal into extreme habitats, it is challenging to comprehend the full extent of the uncharted territory within the ‘dark matter’. However, a mere catalog is insufficient; investigation of this domain offers the potential for discovering novel functionalities that may drive advancements in biotechnology. On a more theoretical level, such exploration has the capacity to provide profound insights into the evolutionary history of *Archaea* and illuminate the origins of complex cellular life, thereby enriching our comprehension of the world [21,22].

In this study, we leveraged a dataset that comprises 500,000 microbial samples, which had been meticulously pre-processed and integrated into the IMNGS database [23]. Our primary objective was to comprehensively investigate the archaeal diversity encompassed within. Specifically, we aimed to accurately determine the lower bounds of this diversity and to pinpoint ecological niches that require further focus. Notably, we shed light on a potential bias in our estimation of the archaeal taxonomic clades within the tree of life. This bias arises from an overrepresentation of cultivable microbes, which leads to an underrepresentation of other taxa. These taxa are routinely discarded despite holding valuable information. We employed a verification process, drawing upon a metagenomic resource for validation, thus ensuring the robustness and reliability of our results. Furthermore, our investigation yielded a significant discovery—the identification of a novel molecular-based class of *Archaea* that resides within the *Asgardarchaeota* clade, which is tentatively denoted as ‘Sleipnirarchaeota’.

## 2. Materials and Methods

### 2.1. Dataset Creation

IMNGS crawls Sequence Read Archive (SRA) [24] and extracts all available 16S rRNA samples regardless of environmental origin or sequencing technology used. At the moment, more than 500,000 processed samples are present in the platform, with 117,000 samples including at least one sequence classified as of archaeal origin. All IMNGS samples have been processed through a pipeline that includes the steps for the following: non-16S-like sequence removal using sortmerna [25], error filtering and Operational Taxonomic Unit (OTU) creation based on the UPARSE [26] algorithm present in USEARCH [27], as well as the taxonomic assignment by RDP [28].

Since every OTU’s assumed taxonomy is already available by the preprocessing pipeline of IMNGS, we extracted the OTUs belonging to the *Archaea* domain from every sample ($n^{seq}$ = 15.8 million). The selected OTUs were compared against the Living Tree Project (LTP) v.128 [29] and SILVA v.138.1 [30] databases, with any exceedingly high (>98%) matched OTU getting replaced by its target (0.2 million hits from LTP and 1.2 million from SILVA). These replacements increase the validity and power of our analysis without reducing its generality due to the databases holding sequences of higher average quality and length and having been expertly verified. Any LTP or SILVA *Archaea* sequences with no hits on our dataset were also appended to our dataset due to them representing archaeal biodiversity not originally present in our dataset.

An important challenge encountered was that the dataset’s OTUs originate from different amplicon studies with different primers used, and the targeted regions are, in the best case, only partially overlapping. To address this challenge, we followed the standard of identifying a common region across studies and trimming all OTUs accordingly, as described in TIC [31] (Appendix A). Since TIC uses the latest SINA aligner in combination with SILVA as the reference database (v138.2), the taxonomy classification of all OTUs was updated. The reclassification process allowed for assigned taxonomies to be corrected, enabling the detection of sequences classified erroneously as *Archaea*, which were discarded ($n^{seq}$ = 12 million).

The extraction of the most-represented region from each OTU and the following collapse of the alignment gaps is a straightforward and easy-to-implement procedure as described in the TIC Pipeline (SINA positions: 10300-25300) (Appendix A). The *Escherichia coli* 16S rRNA gene was also aligned through SILVA, and the number of bases (n = 244) within the selected region was used as the lowest limit of required information each *Archaea* OTU must have for it to be included into the next stage ($n^{seq}$ = 9 million). A dereplication was performed on the extracted sub-sequences with the output acting as our final dataset ($n^{seq}$ = 6.3 million) (Figure 1).

### 2.2. Clustering Limits Identification

In the *Archaea* domain, no well-defined limits exist for the sequence percentage similarity between families, genera, and species, which are required for usage of our clustering tool TIC for the identification of novel molecular families, genera, and species. Even if they existed, those limits would not be applicable to our project since we have extracted a non-standard region of the 16S rRNA gene combining the V4 and V5 16S rRNA variable regions (Appendix A). To overcome this hurdle, we queried all SILVA sequences against those included in LTP. Any match over 98% was appended to LTP and taxonomically characterized by its match. We split this combined LTP/SILVA dataset based on its taxonomic information and calculated the intra-group differences on all levels on the full 16S region. Since our *Archaea* dataset contains a sub-region of 16S, we extracted the same region from the combined LTP/SILVA dataset, keeping the full taxonomic information, so we can split the dataset based on it and re-calculate the intra-group similarities. Through this process, we verified that our selected region almost perfectly mirrors the percentage dissimilarity of the full sequences (family level: 89%, genus level 93%, species level 97%), which verifies that our region is representative of the diversity contained within the full 16S rRNA sequence (Figure 2).

### 2.3. Verification

Verification was performed using Integrated Microbial Genomes and Microbiomes (IMG/M) [32] as a search target. IMG/M contains annotated metagenomes from the three domains of life, which are sequenced at DOE’s Joint Genome Institute, submitted by external users, or imported from the same source as IMNGS (SRA). IMG/M hosts approximately 65 billion metagenome genes processed through the GOLD pipeline [33], which encompasses filtering, error correction, and assembly of reads, followed by annotating the structure and function of contigs and contig-based binning. The volume of IMG/M enables the verification of our novel molecular species, ensuring both their presence in amplicon and metagenomic samples and that they are not artifacts.

## 3. Results

### 3.1. Archaea Knowledge Expansion

GAD was reorganized into 2.8 million species OTUs (SOTUs), 561 thousand genus OTUs (GOTUs), and 98 thousand family OTUs (FOTUs), which are orders of magnitude larger than those contained in the LTP, SILVA, and GTDB [34,35] databases (Table 1, Figure 3).

As expected, the majority of SOTUs, GOTUs, and FOTUs contain a single OTU from only one SRA sample, and removing those singletons decreases the numbers significantly (Table 1).

Another approach in finding the low limit for the global archaeal diversity is based on their presence in different samples. To do this, we counted only the most abundant SOTU from every GOTU for each sample, i.e., when two SOTUs come from the same sample and have been identified as belonging to the same GOTU, only the most abundant SOTU would be counted. This helps account for noise (batch effect) or errors (sequencing limitations) in our data since GOTUs are dissimilar enough to tolerate those differences. Generalizing this for every SOTU, GOTU, and sample, we obtained a low estimation of 1.5 million SOTUs. (Figure 4).

At the family level, the analysis is more complex. When we used VSEARCH [36] to cluster LTP and SILVA sequences from the same family, we obtained multiple clusters rather than a single one. This was expected because taxonomy classifiers like SINA consider factors beyond sequence similarity to assign taxonomy, while naive clustering tools like VSEARCH rely purely on sequence similarity. Since TIC uses VSEARCH for clustering, the reported number of FOTUs is likely an overestimate.

Furthermore, not all taxonomic branches are equal in terms of novelty included (Figure 3). The order *Woesearchaeales* has not been explored enough to identify families, genera, and species mainly due to difficulties in lab cultivation. Our analysis shows that *Woesearchaeales* contains almost half of the diversity in the family and genera levels, and since a third of our dataset’s OTUs (2.2/6.4 million) were taxonomically classified to it, painting it as a valid target for further work (Appendix A). On the other hand, there is evidence of a new narrow order (sequence to species rate = 0.2) within the *Thermoplasmatota* class that was not previously identified.

We have also verified the environmental preferences of *Archaea* being soil and saline waters as reported in [37,38,39,40,41]. Furthermore, the least explored *Archaea* phyla (as evidenced by the number of known classes and orders enclosed) incorporate most of the novelty found, giving us a clear indication of where we should focus our efforts for the purpose of exploration and acquiring new knowledge (Figure 5).

### 3.2. Novelty

A novelty score, which was based on the number of FOTUs, GOTUs, and SOTUs within the samples of each environment, was normalized by the number of environment samples present in our dataset. Saline water samples hold the most unexplored novelty on each level, while host-associated (Appendix A) and plant samples are the most studied.

The sampling effort is not spread equally between all environments; for example, our dataset contains 15 thousand samples originating from saline water with an equal number of host-associated samples despite the fact that 70% of our planet is covered by oceans. However, the rate of identification of new molecular species ($d_{sotu}$) after incorporation of new samples into our data varies greatly, with soil ($d_{sotu}$ > 27) and saline water ($d_{sotu}$ > 25) environments having a lot of diversity that remains unexplored. In contrast, we have to expend a lot of additional effort to find novelty in host-associated and plant environments (Figure 6 and Figure 7).

### 3.3. Verification

IMG/M is a data management system containing metagenomes from a large diversity of microbiomes. We re-categorized the SOTUs to look into their contributing SRA sample. An SOTU containing a single OTU from an SRA sample was termed a singleton whereas a doubleton contains at least one sequence from two different samples, followed by tripletons and moretons. We searched for high-quality matches between GAD and IMG/M. Even though our sequences originated from amplicon studies and IMG/M contains only metagenomes, 94% of our SOTUs were matched up to the family level (89% similarity). (Figure 8). This provides an independent line of support for the validity of the predicted *Archaea* diversity.

Metagenomic sequencing, due to the distribution of the reads across the multiple genomes of the targeted microbial samples, tends to capture predominantly the 16S genes from the most dominant species. Therefore, a match between an SOTU from the GAD collection and the metagenomic sequences of IMG/M would not only provide additional evidence of its presence in the WGS microbiomes but also support its key role in the detected environment. It is expected that cosmopolitan archaeal species would be overrepresented in the metagenomic database, while more niche-specific species would be less represented or not represented at all. Our division of predicted SOTUs from singletons to moretons likely reflects a ranking of ecological spread despite the sampling biases in the databases. Our results show that the SOTU singletons are harder to be found in IMG/M, with only 0.6% of the 2.4 million found to have a near exact match. This rate of matches to the IMG/M sequences grows to 5% for doubleton SOTUs, 7% for tripletons, and 18% for moretons. Nevertheless, moving to a species level of similarity (97%) as a cutoff for matching sequences, only singletons show a distinguishably lower rate of 10% matching while all other sequences were matched at around 30% rate. For higher taxonomic levels (genus/family), the discrepancy in the matching rate disappears (Table 2).

### 3.4. Asgardarchaeota Case Study

*Asgardarchaeota* represent the closest relatives of Eukaryotes [2,42,43,44] from the other domains of life through the existence of genes encoding homologous proteins [45] and other characteristics [46,47]. However, the inability to perform lab cultivation still hampers our understanding of this important phylum. This obstacle is evidenced both by LPSN [48], which contains only three candidate classes, and SILVA containing only three classes (*Heimdalarchaeota*, *Lokiarchaeota* and *Odinarchaeota*) out of the twenty proposed by other studies [13,49,50] at the time of this analysis. Since our approach is based on SILVA as a taxonomic reference, we limited the initial specificity to the three classes only, with all others lumped into the UNKCLASS label.

Based on the known Asgard classes, *Lokiarchaeota* encompass the overwhelming majority of our data (Figure 9a), and each class also shows a difference in its niche environment (Appendix A). From a bird’s eye view, the Asgard SOTUs originate from diverse environments (Figure 9b) dominated by the soil and saline water categories.

All Asgard SOTUs with unidentified class (UNKCLASS) were placed on the phylogenetic tree produced by Liu et al. [49], using EPA-ng [51] (Figure 10).

While no unknown SOTUs were identified as belonging to certain proposed classes (Gerd, Wukong, Heimdal, and Baldr), some were assigned to known classes, reducing the number of unknowns. There are five clusters produced from exclusively novel SOTUs that should be studied more since they may represent novel Asgard classes. Among those, Cluser 5 is the most diverse and encompasses twenty-six SOTUs (six verified by different targets of IMG/M) originating from different environments and twenty-five SRA samples. In combination with the cluster distance from other known classes, we are confident that this is a new class, and we propose the new class name: Sleipnirarchaeota. Based on the samples analyzed in this study, Sleipnirarchaeota are predominantly associated with soil and saline water environments.

## 4. Discussion

### 4.1. Lower Boundary of Archaea Diversity

The extent of archaeal diversity remains a subject of active scientific debate, with estimates ranging from a few thousand to millions of taxa [20]. In this study, we focused on uncovering the archaeal dark matter, defined as archaeal taxa detected in environmental samples originally characterized using methodologies optimized for bacterial identification rather than archaeal-specific approaches. These archaeal populations are often underrepresented due to biases in detection methods, such as primer specificity and limitations in reference databases. However, this study underscores that significant novelty resides within already analyzed and published datasets. It demonstrates how the systematic reanalysis of existing data can yield valuable insights and drive novel breakthroughs, ultimately advancing scientific understanding. By extension, it highlights the importance of the scientific community adhering to the principles of FAIR (Findable, Accessible, Interoperable, and Reusable) data management [52] that enable such analysis. The realization of the hidden wealth available in public repositories has led to the development of specialized algorithms and pipelines for the utilization of available data. The general methodology employed in this study has been validated in prior [53], which refined on the foundational concepts originally proposed by Lagkouvardos et al. [54]. Following this process, we identified a robust lower boundary for archaeal diversity, quantified at 2,807,013 SOTUs. This estimate is supported by evidence from both metagenomic data and curated database records.

Our findings provide a foundational estimate for the lower limit of archaeal diversity and underscore the critical need for archaeal-specific detection and classification methodologies. Addressing these methodological challenges will enable a more comprehensive resolution of archaeal complexity and improve our understanding of their ecological and evolutionary significance within microbial communities.

### 4.2. Sleipnirarchaeota

The first sequences of the superphylum *Asgardarchaeota* were isolated from the sediments around the hydrothermal vents in Loki’s castle in the Atlantic Ocean [55] and consequently classified in the *Lokiarchaeota* class. Initial analysis placed them in a monophyletic relationship with *Eukaryotes*, evidencing a shared common ancestor, which indicates that eukaryotic cells evolved from *Archaea*. A further support to this claim is the presence of homologous genes between them [45,55,56,57,58,59,60].

*Thorarcheota* were the second class to be discovered in estuary sediments, followed by *Heimdallarchaeota* and *Odinarchaeota* [47,55] encountered in anaerobic sediments from hot springs and groundwater [44], *Helarchaeota* were isolated by sampling deep-sea hydrothermal vents [46]. Their late discovery was due to incompatibility with the common PCR primers and their low abundance (>1%).

In recent years, the Asgard phylum has undergone significant expansion, now encompassing at least eleven classes, which challenges previous estimates of *Asgardarchaeota* diversity and raises the intriguing question of whether Eukaryotes represent a deep clade within the Asgard phylum [49]. In this study, we provide evidence supporting the existence of multiple novel molecular classes within the *Asgardarchaeota*, each with varying numbers of members. Specifically, the verification of Cluster 5 from IMG/M strongly supports its classification as a new class. Consistent with the Norse mythology-inspired nomenclature, we propose the name ‘Sleipnirarchaeota’ for this newly identified lineage.

## 5. Conclusions

Herein, we demonstrate that integrative data from high throughput molecular methods allow us to bypass cultivation constraints in the investigation of microbial diversity. The evidence shown here supports the existence of millions of archaeal ‘molecular species’ as well as the presence of several so far unknown higher lineages. Nevertheless, other than giving us some general ecological findings, it is clear that simple amplicon-based data are not sufficient to give insight into the physiology, overall function, and ecology of this massive microbial dark matter. A combination of metagenomic and single-cell sequencing is the natural next step in our quest for understanding our microbial world.

Furthermore, this study reaffirms that our costly, already-published data remain underutilized. Most raw sequence data are deposited in the SRA but see little to no use for contextualization or integration despite the potential demonstrated. Enhancing accompanying metadata would significantly increase their utility and, in combination with specialized overlay tools, could unlock their full potential.

## Figures and Tables

**Figure 1 microorganisms-13-00598-f001:**
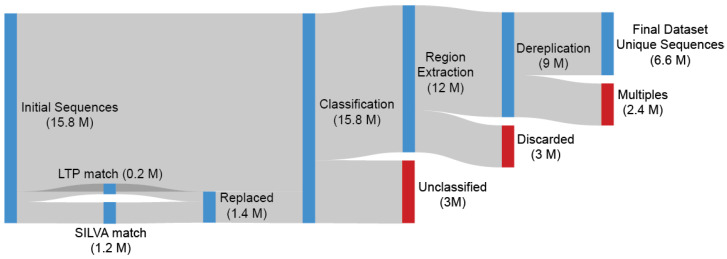
Depicted are 15.8 million *Archaea* sequences originating from 177K SRA samples preprocessed by the IMNGS. Every OTU sequence was searched against LTP and SILVA, with any match over 98% being replaced by the database sequence due to their higher quality. SINA was used for alignment and classification. After identification of the most represented region, *Escherichia coli* 16S rRNA gene was also aligned. The number of *E. coli* bases (n = 244) within the selected region was used as the lowest limit of required information that each *Archaea* sequence included in our dataset must contain, so the sequence is included in the next stage. A dereplication was performed on the extracted sub-sequences with the output acting as our final dataset.

**Figure 2 microorganisms-13-00598-f002:**
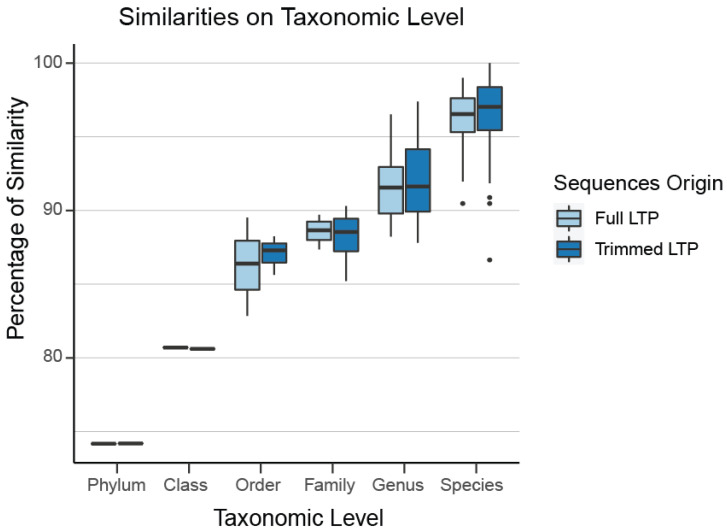
The selected region almost perfectly mirrors the percentage dissimilarity of the full sequences (family level: 89%, genus level 93%, species level 97%). Thus, our region is representative of the diversity contained within the full 16S rRNA.

**Figure 3 microorganisms-13-00598-f003:**
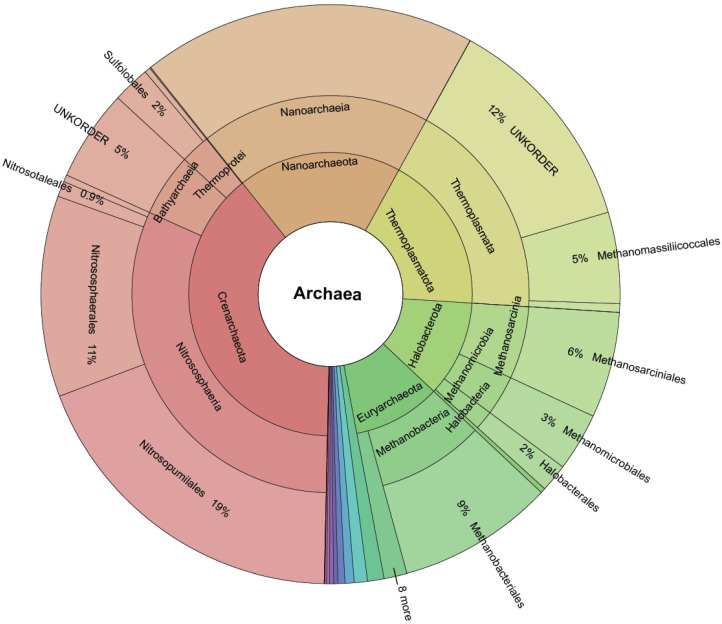
Krona plot quantifying the size of each order after TIC. Novel (created by TIC) and known molecular orders (provided by the SINA classifier) are included.

**Figure 4 microorganisms-13-00598-f004:**
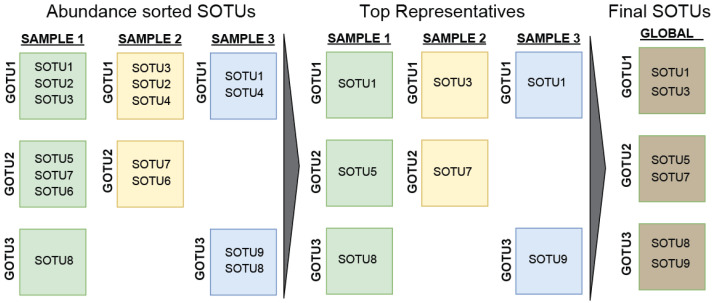
Schematic of the process for estimation of the low limit of archaeal diversity. A list of highly confident SOTUs was assembled by joining the selections of the most abundant SOTU per GOTU per sample for all samples.

**Figure 5 microorganisms-13-00598-f005:**
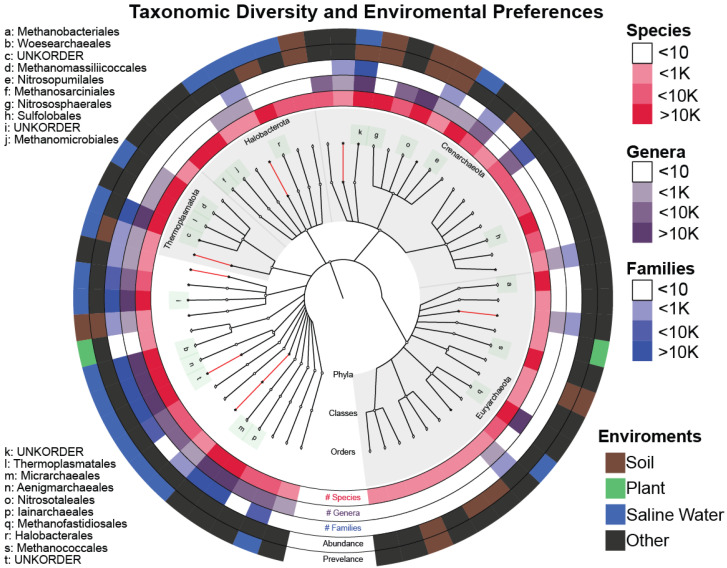
Graphlan plot with the center depicting the taxonomic tree of the SOTUs after TIC that incorporates both novel (red) and known (white) clades up to the order level. The three inner rings quantify the number of (#) SOTUs, genera, and families within each order. Names of the orders containing more than 10 K SOTUs are also given on the left side. The fourth ring (Abundance) shows the environment of the original IMNGS sample, where the most abundant sequence contained within this order comes from. The outer ring (Prevalence) depicts the majority-rule-voted environment from all the sequences contained within the selected order.

**Figure 6 microorganisms-13-00598-f006:**
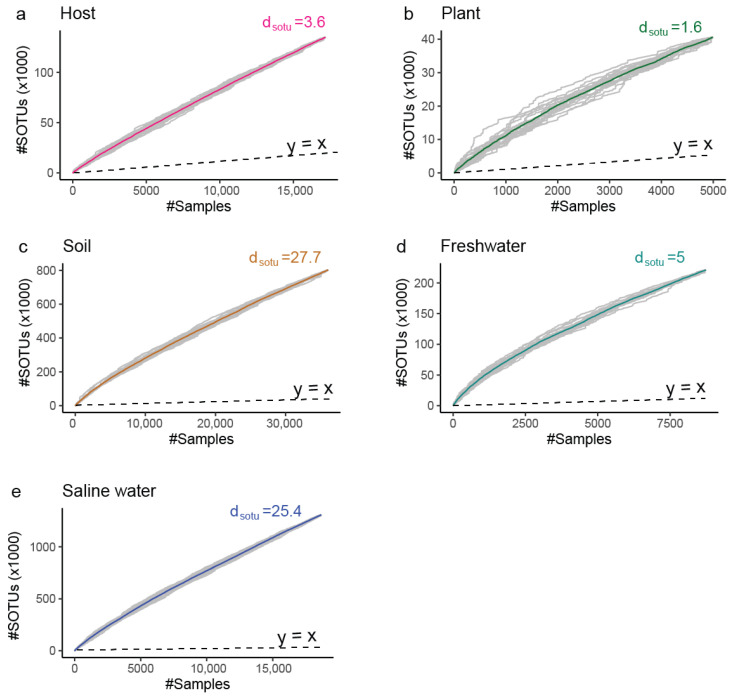
Rarefaction curves indicating the cumulative archaeal diversity at broad ecological niches (gamma diversity).The X axis represents the number of (#) microbial profiles intergrated. The Y axis represents the number of (#) thousands SOTUs discovered up to the selected number of profiles. d_sotu_ corresponds to the expected novel SOTUs by the integration of one additional sample of that niche past those already included in the study. Dashed line is the 1:1 relation rate (1 new SOTU per 1 additional sample). (**a**) Host (Appendix A); (**b**) Plant; (**c**) Soil; (**d**) Freshwater; (**e**) Saline water.

**Figure 7 microorganisms-13-00598-f007:**
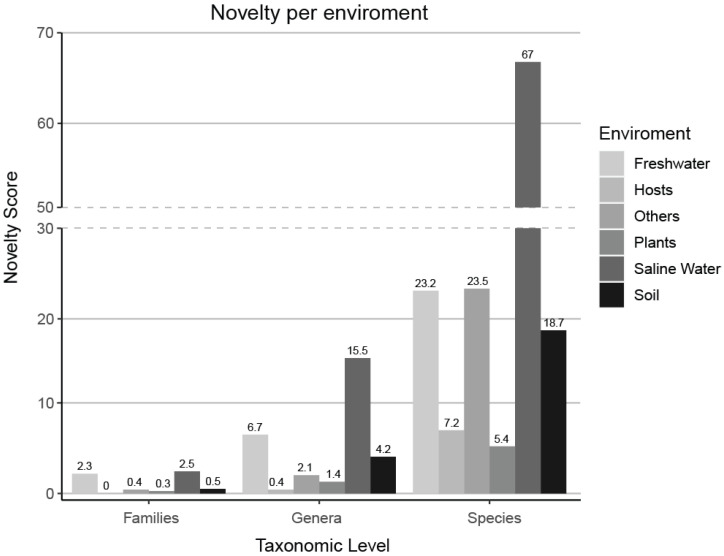
A novelty score, which was based on the number of FOTUs, GOTUs, and sOTUs within the samples of each environment, was normalized by the number of environment samples present in our dataset. Saline water samples contain the highest levels of unexplored novelty across all taxonomic levels, while host-associated samples (Appendix A) and plant samples are the most extensively studied.

**Figure 8 microorganisms-13-00598-f008:**
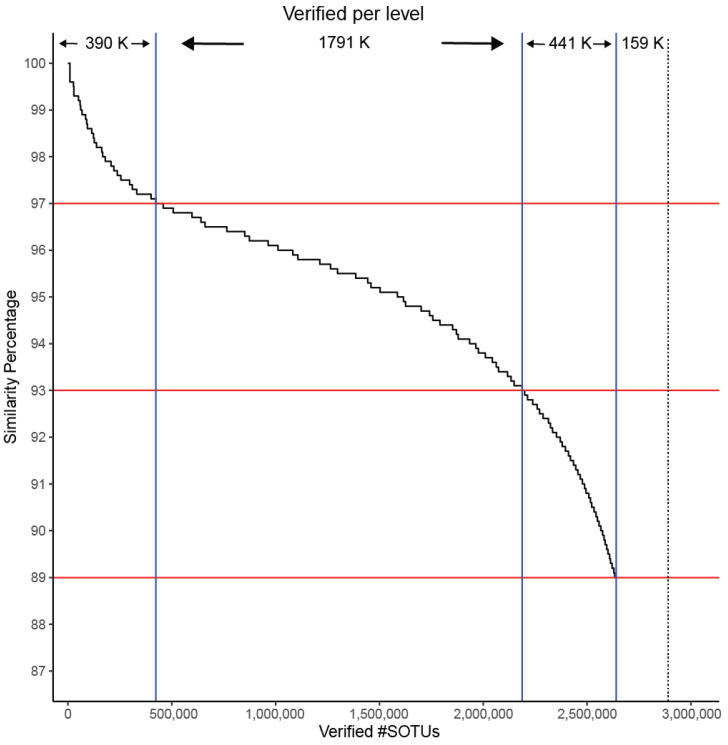
SOTUs verification as a factor of matching to sequences in IMG/M at different similarity levels. Horizontal red lines correspond to similarity cutoffs used for assigning sequences to species (97%), genera (93%), and families (89%). Vertical blue lines correspond to how many SOTUs are verified at each level. Dashed black line indicates the total number of SOTUs.

**Figure 9 microorganisms-13-00598-f009:**
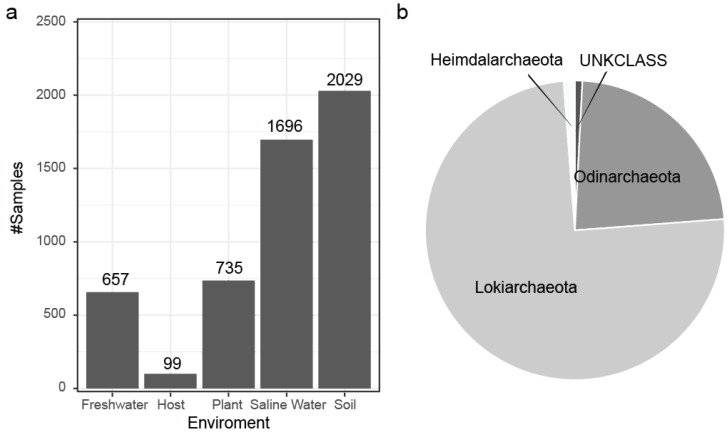
(**a**) Environmental association for the Asgard *Archaea* SOTUs based on the origin of the IMNGS samples with soil being the most rich environment. (**b**) Distribution of the predicted SOTUs to the Asgard classes present in SILVA at the time of this analysis.

**Figure 10 microorganisms-13-00598-f010:**
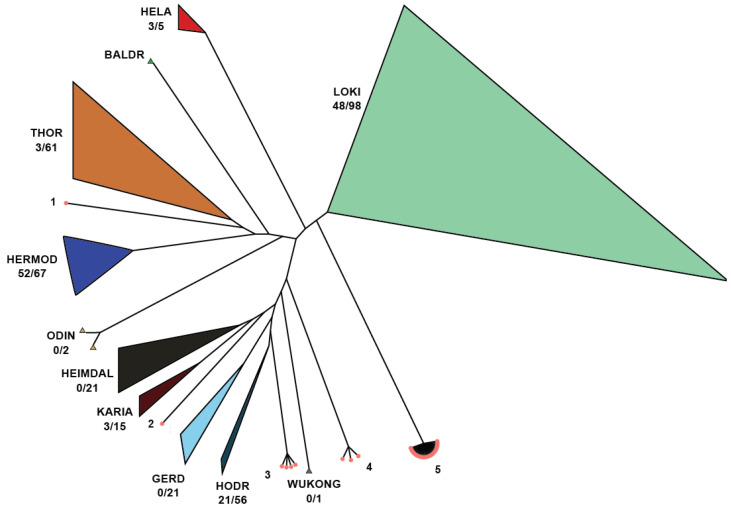
Asgard SOTUs with UNKCLASS placement on the phylogenetic tree. There are five clusters that remain unknown and should be studied more as they may represent novel Asgard classes. Cluster 5 contains 26 SOTUs (6 verified by different targets of IMG/M) from diverse environments and originating from 25 SRA samples.

**Table 1 microorganisms-13-00598-t001:** Comparison between the number of families, genera, and species included in the Living Tree Project (LTP), SILVA, and GTDB databases at the time of our study and those predicted from this study. (# symbol refers to “the number of” the corresponding taxon.)

Database	#Families	#Genera	#Species
LTP	35	129	490
SILVA	82	161	-
GTDB	2621	2769	3044
GAD (singletons in)	98,172	561,788	2,807,013
GAD (singletons out)	30,887	87,200	419,934

**Table 2 microorganisms-13-00598-t002:** Number of SOTUs with match at IMG/M at different similarity levels. SOTUs are divided according to the number of samples positive for their presence.

	SOTUs	99	97	93	89
Singletons	2,398,351	17,986 (0.7%)	254,283 (10%)	1,485,595 (61%)	2,246,610 (93%)
Doubletons	151,600	8157 (5%)	52,428 (34%)	112,662 (74%)	144,255 (95%)
Tripletons	61,824	4905 (7%)	22,172 (35%)	41,752 (67%)	58,978 (95%)
Moretons	195,238	36,579 (18%)	61,631 (31%)	151,397 (77%)	189,018 (96%)
Total	2,807,013	67,627 (2%)	390,514 (13%)	1,791,406 (63%)	2,638,461 (93%)

## Data Availability

The data are available on Github (accessed on 4 March 2025).

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
