# Peer review of "Global Archaeal Diversity Revealed Through Massive Data Integration: Uncovering Just Tip of Iceberg"

_microorganisms, 2025, doi:10.3390/microorganisms13030598_

Round 1
Reviewer 1 Report
Comments and Suggestions for Authors
Antonios Kioukis and co-authors in the MS titled " Global archaeal diversity revealed through massive data integration: uncovering just the tip of the iceberg," processed a large data set, including 500,000 samples of microorganisms, to comprehensively study the diversity of archaea contained in it. In addition, the authors tried to assess the potential error in estimating the taxonomic groups of archaea in the tree of life. The presented material fully corresponds to the profile of the journal Microorganisms and, without a doubt, is of interest to a wide range of researchers.
There are only a few minor points.
Line 120 and Legend for Fig. 1 in “Escherichia Coli 16S” coli should not be started with capital letter.
For Figure 8., provide some suitable title. In its present form, this is not a title, but a piece of text that can also be found in the article's test.
The same, for Table 2 “Table 2. sOTU matching rates were based on the number of samples contributing to them. As expected, lowering the identity threshold from strain-level to species, genus, and family levels increased the number of verified sOTUs.” There is no title, but some explanation.
Line 225-227 “Therefore, a match of an sOTU from the GAD collection to the metagenomic sequences of IMG/M would not only be additional evidence of presence in the WGS microbiomes but also of their key role in the environment detected” I'm sorry, I don't understand why the word "detected" is at the end of the sentence.
Author Response
We first want to express our gratitude to the reviewer for his time and the thorough examination of our manuscript. We also want to apologize for typos and formatting mistakes that slipped our attention. Those errors have been corrected.
We agree with the reviewer's notice that figure and table legends should assist in understanding the depiction or the content of the figure or table. We modify the legends/ titles for figures 4, 6 and 8 and table 2.
Below are the old and new text for them.
OLD -Figure 4. Novelty score based on the number of fOTUs, gOTUs and sOTUs within the samples of each environment normalized by the number of environment samples present in our dataset. Saline water samples hold the most unexplored novelty on each level, while host-associated and plant samples are the most studied.
NEW-Figure 4. Schematic of the process for estimation of the low limit of archaeal diversity. A list of highly confident SOTUs was assembled by joining the selections of the most abundant SOTU per GOTU per sample for all samples.
OLD-Figure 6. Sampling effort is not spread equally among all environments, for example our dataset contains 15K samples originating from saline water with an equal amount of host-associated samples despite the fact that 70% of our planet is covered by oceans. However, the rate of identifying novel molecular species (d sotu) following the incorporation of new samples in our data varies greatly, with soil (d sotu <27) and saline water (d sotu <25) environments having a lot of still unexplored diversity. In contrast, we have to expend a lot of additional effort to find novelty in host-associated and plant environment.
NEW-Figure 6. Rarefaction curves, indicating the cumulative archaeal diversity at broad ecological niches (gamma diversity). d sotu corresponds to the expected novel SOTUs by the integration of one additional sample of that niche past those already included in the study. Dashed line is the 1:1 relation rate (1 new SOTU per 1 additional sample). a) Host (mostly human or rodent), b) Plant, c) Soil, d) Freshwater, e) Saline water.
OLD-Figure 8. Verification of sOTUs by matches in IMG/M. Sorted similarity of all 2.6 Million sOTUs with a match against the total volume of IMG/M over 89% lowest identity and query coverage above 90%. Horizontal red lines correspond to similarity cutoffs used for assigning sequences to species (97%), genera (93%) and families (89%). Vertical blue lines correspond to how many sOTUs are verified at each level
NEW- Figure 8. Plot of SOTUs verification as a factor of matching to sequences in IMG/M at different similarity levels. Horizontal red lines correspond to similarity cutoffs used for assigning sequences to species (97%), genera (93%) and families (89%). Vertical blue lines correspond to how many sOTUs are verified at each level.
OLD-Table 2. sOTU matching rates were based on the number of samples contributing to them. As expected, lowering the identity threshold from strain-level to species, genus, and family levels increased the number of verified sOTUs.
NEW-Table 2. Number of SOTUs with match at IMG/M at different similarity levels. SOTU are divided according to the number of samples positive for their presence.
Reviewer 2 Report
Comments and Suggestions for Authors
The manuscript entitled “Global archaeal diversity revealed through massive data integration: uncovering just the tip of the iceberg” submitted to Microbiology covers very well-designed and interesting data about Archaea diversity. The study involved a modern bioinformatics approach, and the results are presented in high-quality figures and tables. Thus, I have only a few minor editorial concerns:
- Archaea should be written in Italic. Thus, please correct this word in the following lines: 14, 19, 22, the Figure 9 caption.
- Keywords: Archaea – Italic and capital letter; add ‘genetic’ before ‘diversity’.
- Line 121: change ‘Escherichia Coli’ into “Escherichia coli’.
- What was the size of 16S rRNA fragment used in comparison? – provide in section 2.1.
- Figure 6 and Figure 7: please provide the kind of hosts in panel a (Figure 6) and in the legend (Figure 7).
- Lines 228, 319, and 320 change ‘Archaeal’ into ‘archaeal’.
- Conclusion should provide the most information achievements and should not be a repetition of the text. Thus, please remove the first paragraph (The first Archaea that were studied originated from thermophilic environments or were methanogens, resulting in limiting our view of their role as binary, Archaea were either involved in methanogenesis or sulfur respiration [61]. Another reason for our constrained view was the difficulty in cultivating most of the Archaea in the lab, with only Euryarchaeota and Crenarchaeota extensively cultured and studied.) and the third one (The main goal of this paper, however, was to provide a map quantifying the richness in novelty of different ecological niches. This was carried out to help the community explore the novelty in selecting the next sampling niche in a more targeted and educated way).
- Special attention should be given to punctuation. For instance in the following paragraph probably four commas are missing” ‘Verification was performed using Integrated Microbial Genomes and Microbiomes (IMG/M) [32] as a search target. IMG/M contains annotated metagenomes from the three domains of life COMMA which are sequenced at DOE’s Joint Genome Institute, submitted by external users COMMA or imported from the same source as IMNGS (SRA). IMG/M hosts approximately 65 billion metagenome genes processed through the GOLD pipeline [33] COMMA, which encompasses filtering, error correction, and assembly of reads, followed by annotating the structure and function of contigs and contig-based binning. The volume of IMG/M enables the verification of our novel molecular species ensuring both COMMA their presence in amplicon and metagenomic samples and them not being artifacts.’
- Archaea should be written in Italic through the whole text.
- Keywords: Archaea – Italic and capital letter; add ‘genetic’ before ‘diversity’.
- Line 121: change ‘Escherichia Coli’ into “Escherichia coli’.
- Lines 228, 319, and 320 change ‘Archaeal’ into ‘archaeal’.
- Special attention should be given to punctuation. For instance in the following paragraph probably four commas are missing” ‘Verification was performed using Integrated Microbial Genomes and Microbiomes (IMG/M) [32] as a search target. IMG/M contains annotated metagenomes from the three domains of life COMMA which are sequenced at DOE’s Joint Genome Institute, submitted by external users COMMA or imported from the same source as IMNGS (SRA). IMG/M hosts approximately 65 billion metagenome genes processed through the GOLD pipeline [33] COMMA, which encompasses filtering, error correction, and assembly of reads, followed by annotating the structure and function of contigs and contig-based binning. The volume of IMG/M enables the verification of our novel molecular species ensuring both COMMA their presence in amplicon and metagenomic samples and them not being artifacts.’
Author Response
We first want to express our gratitude to the reviewer for his time and the thorough examination of our manuscript. We also want to apologize for typos and formatting mistakes that slipped our attention.
Points 1,2 and 3: Corrected
Point 4: Added in line 122 of the revised manuscript
Point 5: Since there are dozens of different hosts (also sometimes unknown) we cannot add a legend in the figures with details about them. We included a table in the Supplementary data listing the types of sources (description extracted from SRA) and their number for all samples grouped under “Host-associated”. We refer to that table in Figures 6 and 7.
Point 6: Corrected
Point 7: We agree with the reviewer and modified the Conclusion as follows:
Hereby, we demonstrate that integrative data from high throughput molecular methods, allow us to bypass cultivation constraints in the investigation of microbial diversity. The evidence shown here supports the existence of millions of archaeal “molecular species” and also the presence of several so far unknown higher lineages. Nevertheless, besides some general ecological findings, it is clear that simple amplicon-based data are not sufficient to give insight into the physiology and overall function and ecology of this massive microbial dark matter. A combination of metagenomic and single-cell sequencing are the natural next steps in our quest for understanding our microbial world.
Furthermore, this study reaffirms that our costly, already-published data remain underutilized. Most raw sequence data are deposited in the SRA but see little to no use for contextualization or integration, despite the potential demonstrated here. Enhancing accompanying metadata would significantly increase their utility, and in combination with specialized overlay tools, could unlock their full potential.
General comment: We thoroughly examined the manuscript and corrected for missing punctuation.